# The impact of digital financial development on corporate leverage ratio: The case of a-share listed non-financial enterprises in China's Shanghai and Shenzhen stock exchanges

Liu Junqi[1], Sher Abbas[1] *, Liu Rongbing[1] *, Najabat Ali[2,3]

1 School of Finance and Trade, Liaoning University, Shenyang, China, 2 Faculty of Management Sciences, Hamdard University, Islamabad, Pakistan, 3 School of Business, Soochow University, Suzhou, China

* soleun@163.com (LR); abbassher553@gmail.com (SA)

**Data Availability Statement:** All relevant data are within the paper and its Supporting Information files.

## Abstract

This study investigates the impact of digital finance on corporate leverage ratios. The study employed a large sample of China's Shanghai and Shenzhen A-share non-financial listed enterprises from 2011–2020. The study's results depict that the development of digital finance can significantly reduce the leverage ratio of enterprises. We empirically identified that digital finance affects the difference in the term structure of the corporate leverage ratio. It was found that the development of digital finance has a significant negative impact on enterprises' short-term and long-term leverage ratios. Moreover, our heterogeneity analysis shows that the negative effect of digital financial development on corporate leverage ratios is different in state-owned and non-state-owned enterprises, large-scale and small-scale enterprises, and high-leverage and low-leverage enterprises. Mechanism analysis shows that the development of digital finance can reduce corporate leverage by lowering financing costs, alleviating financing constraints, and weakening non-systemic risks. Therefore, policymakers should focus on developing and adopting digital finance by creating a supportive regulatory environment, improving access to digital financial services, and encouraging innovation in the digital finance sector. Finally, our results remain robust after addressing endogeneity issues and conducting robustness checks.

## 1. Introduction

Global economic and financial development has recently become more advanced and assistive [1]. Factors of instability and financial uncertainty are gradually increasing, and sustainable financial development has gained broad perception [2]. Since the last quarter of 2022, China's government has improved COVID-19 prevention and control financial policies, which leads to control market expectation of a regional economy, which recovered from the substantial contradiction between "asset shortage" and "capital shortage", resulting in a mismatching "financial supply and financial demand [3]. This has increased the firm's leverage ratios;

**Funding:** The author(s) received no specific funding for this work.

**Competing interests:** The authors have declared that no competing interests exist.

according to the NIFD (National Institute of Financial Development report, the leverage ratios of non-financial China's corporate sector were 167% in the first quarter of 2023, reaching the highest level globally. Therefore, the corporate financial leverage concept fits the economic and financial development context and has received prevalent attention from researchers and practical sectors [4].

Corporate leverage ratios are financial metrics used to assess a company's debt levels and ability to meet its financial obligations [5]. These ratios are crucial for investors, creditors, and management to gauge a company's financial health and risk exposure [6]. Key corporate leverage ratios include the debt-to-equity ratio, the debt-to-assets ratio, and the interest coverage ratio [7]. For several reasons, digital finance and corporate financial ratios are more significant from a business perspective. The digital finance system has facilitated financing and credit options for small and large enterprises [8]. This increased access to financing can impact corporate leverage ratios, which may lead to companies opting for debt financing to fuel growth and expansion [9]. Secondly, digital finance solutions enable businesses to streamline their financial operations and improve cash flows, potentially leading to improved debt management and reduced leverage ratios [10]. Making real-time financial decisions based on accurate data and automated processes can help businesses maintain a healthier financial position [11]. In addition, digital finance systems can provide advanced financial data to analytics and risk mitigation tools that allow investors and creditors to evaluate the accurate information of a firm's creditworthiness more efficiently and effectively [12]. As a result, firms with financially strong and lower leverage ratios can have better borrowing policies and lower interest rates.

The emergence of new digital technologies such as artificial intelligence, information, big data, machine learning, and cloud computing are globally used in firms' production and management decision-making processes [13]. Digital finance can reduce firms' operational costs by obtaining more accurate financial information. It not only enhances the coverage of financial services but also attracts the corporate financial sector to enjoy them [14]; in addition, the digital finance model is an economic model in which financial institutions and internet-based technology firms use digital technologies to develop activities to improve the financial sector [15].

While the existing studies have investigated "digital finance" mainly on a macro level, such as the impact of digital finance on economic growth [16], digital finance on entrepreneurship [17], digital finance on household consumption [18], and digital finance on urban innovation ability and economic growth [19]. There are relatively few studies on the micro-corporate level. These studies mainly focus on digital finance's impact on corporate innovation, such as the impact of digital technology ESG firm's level performance [20], corporate financial destress [21], corporate financial fraud [4], financial inclusion and stability [22], financial risks [23], and digital finance on household debt ratios [24]. However, no previous studies existed on the impact of digital financial development on corporate leverage ratios. Therefore, this study is meaningful in investigating the effect of digital finance on corporate financial leverage ratios.

This study is based on the A-share listed of non-financial enterprises in Shanghai and Shenzhen stock exchanges from 2011–2020 to examine the impact of digital financial development on the corporate leverage ratio of enterprises to explore the effect, dynamic change process, heterogeneity, and indirect mechanism effect of digital finance development on corporate financial leverage ratios. The results show that digital finance reduces corporate financial leverage performance more pronounced than the state enterprises, with high and low leverage, and reduces the corporate leverage ratio by lowering financing costs, easing financing constraints, and weakening non-systemic risks. In addition, improving the corporate leverage ratio, increasing financial investment, and increasing firm cash flow are essential factors through which digital finance promotes corporate leverage.

The contribution of this research study is as follows: based on previous studies discussed corporate financial leverage, e.g., corporate governance [25], alleviating firms' asymmetry [26]. However, big data artificial intelligence, code of computing and AI as a global new generation of financial technology, the relationship between digital finance and corporate leverage ratios has mainly been ignored by investigating digital finance as a crucial factor that can significantly curb corporate leverage ratios. Therefore, the study contributes to existing literature in several ways. Firstly, it examines the influence of digital finance development on corporate leverage ratios at the micro-enterprise level. It elevates research on digital finance and how to impact insights for effectively reducing enterprise leverage ratios. Second, we attempted to contribute the latest literature that investigates the impact of digital finance on corporate financial leverage performance; specifically, we employ a non-financial enterprise-level analysis. and the role of DIF in affecting firms' financial activities, and extend previous studies from the perspective of financial innovation and sustainability. Whereas previous studies have mainly investigated digital finance's ESG efficiency and improvement [20, 27, 28], digital finance financial inclusion [22, 29], financial fraud [4, 30], and financial distress [21]. In contrast, this study provides valuable concepts on digital finance and enterprise efficiency from the perspective of corporate financial leverage. Third, this study provides more detailed results on the intermediate mechanisms by which digital finance can digitally manage firms' financial leverage ratios to reduce financial risk and constraints. This study contributes to the enterprise-level data to test the mechanism between digital finance and corporate leverage, fulfilling the lack of experimental evidence. Fourth, the mechanism of digital finance development on corporate leverage ratios should provide solutions for preventing and mitigating a series of risks that excessive corporate leverage ratios may cause. Finally, it tests the impact of the external economic environment on the relationship between digital finance development and corporate leverage ratio, which can provide a basis for what measures the government takes to effectively play the positive role of digital finance.

The remainder of the study is organized as follows: Section 2, literature review and research hypotheses. Section 3 discusses the theoretical framework and empirical methodology. Section 4 analyzes the empirical results. Section 5 conducts the mechanism analysis and heterogeneity analysis. Section 6 external economic impact of digital finance. Section 7 conclusions and policy recommendations.

## 2. Literature review and research hypotheses

### 2.1 Literature review

The research on the development of digital finance on corporate leverage ratio has been deepening since Fisher [31] constructed the debt-deflation theory, which has become the focus of discussions in the field of macroeconomics and corporate finance at the macro level. Furthermore, many scholars conducted in their prior studies on the economic effects of digital finance specifically focused on macro and micro-economic perspectives [13, 14]. In addition, some studies empirically analyzed the impact of poverty reduction, such as Li et al [19], and Das and Chatterjee [32]. Digital finance inclusive and digital growth [33]. For example, some researchers have empirically analyzed digital finance in the corporate banking sector [34], corporate innovation [35], financial risk [23], financial stability [36], and financial innovation [22]. From the perspective of business evolution, digital finance development has enhanced competition among the financial sectors [37]. However, digital technology in the financial industry can bring financial risks [23].

From a business perspective, scholars also incorporated the economic benefits of digital finance on financial development into their studies, analysed the correlation between digital

finance and financial inclusion, and identified economic indicators such as digital finance relations and financial stability [38], financial efficiency [39], corporate financial fraud [4], and financial distress [21]. Furthermore, some research scholars significantly clarify the benefits of digital finance in the context of social and environmental perspectives, such as corporate ESG [20, 40] analyzed the benefits of digital finance on carbon emissions and clarified that the impact of digital finance can significantly reduce carbon emissions in specific sectors. However, few studies have been analyzed on microeconomic indicators [41]. Few researches have been conducted on household consumption [18], and few studies have been examined from the perspective of digital finance on corporate innovation [41, 42]. In addition, few scholars address the impact of digital finance from the perspective of corporate financial constraint, efficiency, and performance [28, 43]; beyond rare studies have focused on digital finance on the financial efficiency of firms.

On the other hand, most of the latest studies empirically conducted by different scholars on digital financial development from the perspective of enterprises' financial control, such as Sun et al [4], corporate financial distress [21], enterprises financial deleverage [44], but few research studies have explored the impact factor of corporate leverage [45]. Scholars have clarified that corporate leverage investment can significantly improve corporate value [45]. Operating performance can reduce financing costs, enhance corporate investment performance and promote financial innovation to mitigate financial risk [46], after verifying the important of corporate leverage ratios investment and further including what factors impact corporate leverage performance and how to enhance corporate operating efficiency and achieve financial development by improving corporate leverage ratios performance. This study explores the performance of digital finance and corporate leverage ratios to answer these questions before conducting the empirical analysis. Our proposed research hypotheses are given below. However, limited attention has been paid to identifying the determinants of corporate leverage performance and strategies for enhancing corporate operating efficiency to achieve financial development through improved leverage ratios. Thus, this study aims to fill this gap by examining the relationship between digital finance performance and corporate leverage ratios, thereby shedding light on key drivers of corporate financial health and efficiency in the digital era. This revision aims to provide a more structured and coherent overview of the literature, emphasizing the gap in existing research and the specific focus of the current study. Additionally, it suggests directions for future research by highlighting the importance of understanding the determinants of corporate leverage performance and strategies for enhancing corporate operating efficiency in the context of digital finance.

## 2.2 Relationship of digital Finance with the leverage ratio

The main objective of this study is not only to investigate the relationship between digital finance and corporate leverage ratios but also to address the impact mechanism among them. Dai et al [44] argues that the development of digital finance can reduce the leverage ratio of firms through facilitating of financial constraints and reducing finance cost. On the other hand, corporate leverage ratios play a crucial role in negative moderation. However, the development of digital finance alleviates financing constraints and information about the firm's asymmetry. Existing studies have clarified that the development of digital finance can stress firms' access to finance by using advanced financing services technology, e.g., P2P landing and third-party payment, to meet financial needs [44]. The development of digital finance can connect the large number of similar groups created by traditional finance with the consistent low threshold, low cost, and high-efficiency internet-based financial services to engage them to get liquidity resources, and firms no longer need to obtain leverage funds. In addition, the

development of digital finance can reduce the firm's production and operating costs and improve productivity by using emerging technologies such as artificial intelligence, information, big data, and computing codes [47].

On the other hand, digital financial development is methods of a corresponding digital ecosystem. For example, big data information and credit collection and internet-based technological investment, which enables digital finance to obtain efficient information processing and risk mitigation capacity and significantly reduce the transaction cost, time cost, and employing information technology. Further helping firms to reduce financial costs and obtain funds with higher efficiency, thus achieving the goal of lower leverage [44]. In another hand, digital finance can enhance the efficiency of capital utilization, and also reduce capital destruction risk in financial mechanism process, thus helping enterprises stabilize their financial level, reducing their demand for external leverage financing. In addition, developing digital finance promotes enterprises' digital transformation, improving their management efficiency and financial control ability and avoiding ineffective leverage. In other words, enterprises can obtain funds efficiently, which is reflected in the fact that digital inclusive finance can reduce the leverage level of enterprises by reducing their financing costs.

The development of digital finance can reduce the cost of assessment and audit costs when financial institutions credit enterprises, thereby improving the efficiency of investment and financing to achieve optimal resource allocation. Moreover, after the lending of funds, digital finance can also enable financial institutions to increase the real-time understanding of enterprises through information platforms. For instance the Internet, effectively supervise the flow of their loan funds, ensure the safety of funds, reduce the loss of bad debts, and thus reduce the financing costs of enterprises [48]. In addition, it can be seen from the cost-benefit theory that the reduction of financing costs will reduce the cost of production and operation of enterprises, increase the income of their investment and operation, enable enterprises to have more surplus funds for large-scale reproduction and debt repayment, thereby reducing the leverage ratio of enterprises. From the perspective of capital supply, digital finance can also alleviate the financing constraints on enterprises by making up for the shortcomings of traditional financial services and expanding the coverage of financial services [49]. This is because, on the other hand, digital finance can break through geographical restrictions, realize the diversification of investment entities, and improve the availability of financing funds. In addition, it can enhance the practical identification and selection of the demand side of funds with the help of continuous improvement of information technology, reduce the probability of mismatch of funds, and weaken the degree of financing constraints on enterprises. From the viewpoint of capital demand, the development of digital finance can enable non-state-owned and small-scale enterprises with limited collateral, such as liquidity constraints initially excluded from traditional finance, to obtain more credit funds and reduce their financing constraints [50]. It can be seen from the theory of financing constraints that the reduction of the degree of financing constraints can enable enterprises to obtain the funds needed for investment, production, and operation on time, reduce the financial risks or financial difficulties that enterprises may encounter, and ensure the effective investment needs of enterprises. Furthermore, the effective investment needs of enterprises will help improve their profitability so that they have more surplus funds for debt repayment, thereby reducing corporate leverage.

In addition, agency theory [51] shows that the principal-agent relationship problem caused by information asymmetry will lead to "short-sighted behavior" in management and the incentive to use management rights to speculate about improving their profits, which is not only detrimental to the development of the enterprise but also increases the risk in the business process of the enterprise [52]. The development of digital finance can also improve the flexibility of enterprises in dealing with risks by helping them establish and improve their own risk early

warning systems and control systems [53]. Reducing operational risks and enhancing risk-bearing capacity will help increase the profitability of enterprises and the ability to repay debts, thereby reducing the enterprise leverage ratio. However, combined with the theory of cost and benefit, theory of financing constraints and the theory of agency, the development of digital finance may reduce the leverage ratio of enterprises by lowering financing costs, alleviating financing constraints, and weakening business risks. Because of this, the following research hypothesis is proposed.

H1: The development of digital finance can reduce the corporate leverage ratio of enterprises.

## 2.3 Heterogeneity analysis

**2.3.1 Firms' sizes and leverage level.** The trade-off theory indicates that corporate leverage is determined by the costs of bankruptcy and agency problems, necessitating the incurrence of corporate liabilities for the tax shield benefits [54]. For high-leverage enterprises, debt levels may exceed the reasonable threshold, resulting in high costs. Therefore, the expected bankruptcy costs and agency costs caused by their high debt are higher than those of low-leverage enterprises. Additionally, because the tax shield income has a marginal decreasing effect, the marginal revenue brought by the increase in the debt of high-leverage enterprises is less than that of low-leverage enterprises. Under the influence of these factors, the development of digital finance may impact the leverage ratio of enterprises by optimizing resource allocation and reducing the costs of enterprise bankruptcy and agency problems. As a result, the deleveraging losses and benefits of high-leverage enterprises are smaller than those of low-leverage enterprises. Small-size enterprises' financing conditions and interest charges have been widely studied in the literature [55]. Due to their strong market reputations and access to capital, some researchers have found that large enterprises tend to have better financing conditions and lower interest charges. On the other hand, small-scale enterprises often face more financial constraints and must contend with unfavourable debt conditions and higher interest charges. Therefore, inventors are more confident in large-scale enterprises as their portfolios are more secure and diversified; hence, they have a lower risk of defaults. Nyeadi and Sare [56] documented that different corporate leverage levels significantly affect enterprises profit. Enterprises with high leverage suffer from the higher cost of short-term debt [56, 57]. Such firms mostly use internal sources to repay debt and rely on external debt for working capital [58]. However, lenders have severe liquidity concerns with high-leverage firms because a significant portion of liquid assets is required to pay back debt charges [59]. High-leverage firms already suffer from a heavy debt load, and lenders are unwilling to provide loans with average interest rates, so these firms have to pay premium interest rates on further debt. Also, these firms have a high risk of insolvency and bankruptcy, so that they may have limited investment opportunities. As a result, these firms cannot earn as much profit as low-leverage firms. In addition, different levels of financing discrimination incentivize large-scale firms with lower loan costs to engage in financial markets as financial intermediaries for intermediary benefits, which not only expands the scale of shadow banking but also makes large-scale enterprises more leveraged [60, 61]. The development of digital finance can reduce the degree of financing discrimination faced by small-scale enterprises and stabilize the intermediary income of large-scale enterprises. However, under the influence of these two aspects, the development of digital finance may significantly impact the leverage ratio of large-scale enterprises with higher leverage levels than small and medium-scale enterprises with relatively low leverage levels. This results in digital finance development having a relatively sizeable deleveraging effect on high-leverage enterprises. Based on the above analysis, the following research hypothesis is proposed.

H2: The development of digital finance can reduce corporate leverage ratios, affecting both large-scale and small-scale enterprises as well as high-leverage and low-leverage enterprises.

**2.3.2 State-own and non-state own enterprises.**   Under the traditional financial structure dominated by banks, the allocation of credit funds in China has a bias towards state-owned enterprises, making it easier to obtain credit funds from financial institutions such as banks at a relatively low cost than non-state-owned enterprises, resulting in state-owned enterprises having a higher leverage ratio than non-state-owned enterprises [60]. From the theory of financial intermediation, obtaining loans from financial institutions such as banks at a relatively low cost will incentivize state-owned enterprises to act as financial intermediaries in the financial market to earn intermediary benefits, especially in the case of their low profitability [57]. Non-state-owned and state-owned banks typically operate with different objectives. Non-state-owned enterprises predominantly seek to maximize returns on capital invested by their owners. At the same time, state-owned banks adhere to regulations established by government officials to fulfil overarching state-owned objectives [62, 63]. Due to different aims, goals, and incentive structures, state-owned and non-state-owned banks exhibit different lending behaviors, resulting in differences in loan allocation and pricing strategies. Based on previous research analyzed the relationship between the digital inclusive index and corporate leverage ratio [64]. Their findings showed that digital finance development significantly impacts the corporate leverage ratios of both state-owned and non-state-owned enterprises, according to Luo [59]. Digital finance development significantly influences the corporate leverage ratios of state-owned and non-state-owned enterprises in China, as the empirical evidence demonstrated by Luo [59]. Their findings suggest that the development of digital finance positively reduces corporate leverage ratios in both state-owned and non-state-owned enterprises in China. In addition, based on the research conducted by Liao et al [64], using digital finance technologies can potentially impact the leverage ratios of both Chinese state-owned and non-state-owned enterprises. They argue that digitization in digital finance can increase operational efficiency and consumer-based services, contributing to greater transparency in business management. Additionally, the research by Wu and Huang [43]. suggests that digital finance can significantly reduce stock price crash risk, emphasizing its positive impact on corporate leverage ratios. This changed in 2015 and 2016 when the People's Bank of China responded to the growth of new fintech corporations. On the one hand, developing digital finance can improve the availability of financing funds to reduce the financing costs of non-state-owned enterprises. On the other hand, it can stabilize the intermediary income of state-owned enterprises by reducing information asymmetry, thereby inhibiting the leverage ratio of state-owned and non-state-owned enterprises. Therefore, this paper proposes the following hypothesis.

H3: The development of digital finance can reduce the corporate leverage ratios of both state-owned and non-state-owned enterprises.

## 2.4 Mechanism analysis

**2.4.1 The development of digital finance affects corporate leverage ratios.**   This study's main purpose is to identify the relationship between the development of digital finance and corporate leverage ratios and examine how digital finance reduces corporate leverage by lowering financing costs, alleviating financing constraints, and weakening non-systemic risks. The development of digital finance offers several benefits for businesses, such as reducing financial risk and alleviating financing constraints. Firstly, digital finance provides businesses access to alternative funding sources [10, 63]. These sources include crowdfunding platforms, peer-to-

peer lending platforms, and online marketplaces. By diversifying their sources of financing, businesses can reduce their reliance on traditional financial institutions and increase the chances of securing funding for their operations. Second, digital finance plays a crucial role in alleviating enterprise financing constraints. It provides new financing channels, such as crowd-funding platforms or peer-to-peer lending, which can connect enterprises with potential investors or lenders specifically interested in supporting enterprises [65]. These platforms use digital technologies to facilitate capital matching with enterprises, reducing the barriers to accessing funding. Additionally, digital finance allows enterprises to reach a larger pool of investors or lenders beyond their local industry [66]. This increased access to funding sources expands the opportunities for enterprises to secure the financing they need to grow and develop. Furthermore, digital finance platforms offer faster and more streamlined loan application processes than traditional financial institutions. This speeds up the funding process and reduces the time and effort required for enterprises to obtain financing. In addition, digital finance tools enable enterprises to automate financial processes, such as invoice management and payment collection, reducing the risk of errors or delays. This automation increases financial efficiency and reduces the likelihood of financial mismanagement or fraud, thereby minimizing financial risk for the enterprise [59]. Finally, Digital finance solutions, such as financial management software or dashes, provide real-time data on various financial metrics, allowing enterprises to continuously monitor and assess their financial health. This real-time access to financial data enables timely identification of potential risks and the ability to take proactive measures to mitigate them [67]. This includes identifying cash flow issues, monitoring debt levels, assessing liquidity positions, and evaluating financial ratios [68]. Moreover, digital finance reduces the enterprise's financial risk by alleviating financing constraints, streamlining financial processes, enabling real-time access to critical financial data, and enhancing risk assessment and monitoring capabilities through big data analytics. Therefore, this paper proposes the following hypothesis:

H4: Digital finance can reduce an enterprise's financial risk through alleviating financing constraints. Fig 1 shows the conceptual model.

## 3. Sample selection and data sources

This study derives digital finance data from the Digital Finance Inclusive Index compiled by the Peking University Center for Internet Finance Research [69]. The enterprise-level data primarily originates from CSMAR (China Stock Market & Accounting Research), focusing on China's Shanghai and Shenzhen A-share listed enterprises spanning 2011 to 2020. Firstly, the data will be organized based on heterogeneity analysis. Excluding financial and real estate firm data helps streamline the analysis by focusing on a more specific group. Financial and real estate companies typically have similar financial structures and behaviors to firms in other sectors. For example, financial institutions may have different leverage techniques because they rely on debt for financing operations, while real estate companies may have specific asset valuation methods. Therefore, we excluded these groups, and after that, we created a more identical dataset, making it easier to identify shapes and relationships related to digital financial development and corporate leverage ratios. This focused method improves the clarity and accuracy of our analysis. Secondly, enterprises classified as S.T (Special Treatment) were removed. S.T enterprises are characterized by abnormal financial situations or conditions, such as consecutive years of negative profit, shareholders' equity lower than registered capital, and audit reports with adverse opinions or a disclaimer of view. Thirdly, samples containing missing values were excluded from the dataset. Following these screening steps, a total of

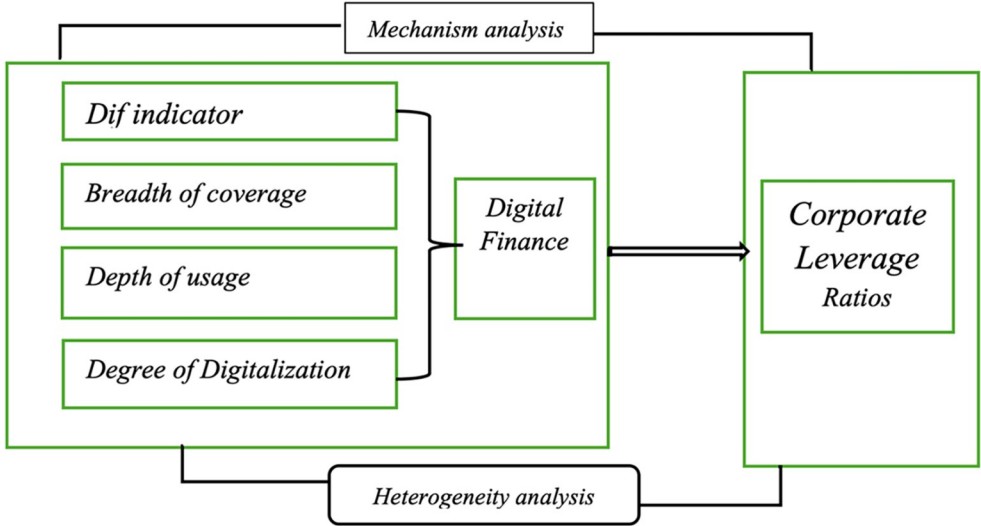

**Fig 1. Conceptual model.** Note: Hypothesis: H1: H1: The development of digital finance can reduce corporate leverage ratio of enterprises. Hypothesis: H2: The development of digital finance can reduce corporate leverage ratios, affecting both large-scale and small-scale enterprises, and high-leverage and low-leverage enterprises. Hypothesis: H3: The development of digital finance can reduce the corporate leverage ratios, of both state-owned and non-state-owned enterprises. Hypothesis: H4: digital finance reduces the enterprise's financial risk by alleviating the financing constraint.

27,739 observations were retained. To mitigate the potential impact of extreme values on empirical results, all continuous variables underwent a top and bottom 1% tail reduction through Winsorization.

## 3.1 Model specification

Based on hypotheses development, in this study, we employ an econometric approach utilizing panel data regression analysis to examine the relationship between digital finance development and leverage ratios. Stata software is used as an analytical tool because of its robust capabilities in handling panel data and estimating regression models. The rationale for selecting this econometric model lies in its ability to control unobserved heterogeneity across entities and periods, providing a more accurate estimation of the relationship between digital finance development and leverage ratios. By including control variables indicating enterprise characteristics, industry effects, and time trends, we aim to investigate the multifaceted nature of the relationship and generate insights into how digital finance impacts leverage ratios among various enterprises. Drawing on the research methods of existing literature [70], the model is specified as follows.

$$lev_{it} = \lambda_0 + \lambda_1 dfi_{it} + \sum_j \varphi_j cv_{jit} + \sum_i \delta_i Industry_i + \sum_t \rho_t Year_t + \varepsilon_{it} \qquad (1)$$

In testing Hypothesis 1, we employed model (1), where lev represents the corporate leverage ratio of enterprise i at year t DIF represent the level of digital financial development, CV encompasses a set of control variables, and Industry and year account for industry and time fixed effects, respectively. The variable ε represents the random error term, and j represents the control variable. Focusing on the regression coefficient $\lambda\_1$ of the digital financial development level (DIF), a positive and significant $\lambda\_1$ would imply that the development of digital finance increases the corporate leverage ratio, leading to the rejection of Hypothesis 1. Conversely, a negative and significant $\lambda\_1$ would support Hypothesis 1, indicating that the

development of digital finance does not increase the corporate leverage ratio. To explore potential heterogeneity in the impact of digital financial development on the corporate leverage ratio based on enterprise characteristics, we extend Model (1) by introducing interaction terms (DIF× var). These interaction terms incorporate digital financial development and enterprise grouping variables, allowing us to examine the size and significance of the coefficients. If the coefficient of the interaction term (DIF × var) is harmful and significant, Hypothesis 2 is validated. Conversely, if the coefficient is not harmful and significant.

## 3.2 Variables definition

**3.2.1 Corporate leverage ratio.** The corporate leverage ratios are measured by the total liabilities divided by the total assets as a proxy indicator used by Kanoujiya et al [71] and Muthama et al [72]. Corporate leverage ratios with different maturities will impact a company's solvency, production, and operation; the corporate leverage ratio is divided into short-term leverage ratio (LEV) and long-term leverage ratio (LLEV) to comprehensively examine the impact of digital financial development. The short-term leverage ratio (SLEV) is measured by the current liability ratio to total assets. In contrast, the long-term leverage ratio (LLEV) is the ratio of non-current liabilities to total assets. Table 1 highlights the concepts and definitions of variables.

**Table 1. Variables concept and definitions.**

| Variable | Abbreviation | Theoretical definition | Operational definition |
|---|---|---|---|
| Corporate leverage | LEV | Measures total corporate leverage | Total Liabilities ÷ Total Assets |
| Short-term leverage | SLEV | Measures the short-term leverage of an enterprise | Current Liabilities ÷ Total Assets |
| Long-term leverage | LLEV | Measures the long-term leverage of an enterprise | Non-current liabilities ÷ Total assets |
| Digital finance development | DIF | Measures the level of digital finance development | The natural logarithm of the digital financial inclusion index |
| Fixed assets proportion | FIX | Measures the fixed assets of an enterprise | Fixed assets ÷ total assets |
| Return on Assets | ROA | Measures an enterprise's asset utilization | Net profit ÷ Total assets |
| Operating cash flow | Cashflow | Measure the liquidity level of an enterprise | Operating cash flow ÷ total assets |
| Revenue growth rate | GRO | Measures the growth capacity of an enterprise | (Current operating income—last period operating income) ÷ last period operating income |
| Tobin's Q | Tobinq | Measure future investment opportunities of a business | Market Cap ÷ Total Assets |
| Equity concentration | Top1 | Measure the governance structure of an enterprise | The sum of the shareholding ratios of the top ten shareholders |
| Enterprise size | Size | Measure the size of a firm | Natural logarithm of total assets |
| Time to market | Age | Measure the longevity of a business | The natural logarithm of (current year—listing year + 1) |
| Broad money growth rate | M2 | Measure changes in monetary policy | Money growth rate |
| Economic growth rate | GDP | Measure changes in the macroeconomic environment | Economic growth rate |
| Industry fixed effect | Industry | Control industry-level differences | Industry dummy variables are set according to the *CSRC's 2012 industry classification standards. Those belonging to the industry take the value of 1, otherwise the value is 0. |
| Time fixed effect | Year | Control the effects of unobservable time factors | Time dummies are set as 1 for year, otherwise 0. |

Note

*CSRC = China Securities Regulatory Commission

**3.2.2 Digital Finance Development (DFI).**   Drawing on existing literature, the natural logarithm of the China Digital Inclusive Finance Index at the provincial level, published by the Digital Finance Research Center of Peking University, is used as a proxy indicator for the development of digital finance [69]. The index is based on digital financial inclusion and the data provided by Ant Group. It is constructed by covering three dimensions: the depth of use of digital finance, the breadth of coverage and the degree of digitalization. Relevant research has proved it to characterize the development level of China's digital finance.

**3.2.3 Control variable (cv).**   Control variables, drawing extensively from existing literature with a focus on both macroeconomics and microeconomics include factors such as fixed assets (Fix), return on assets (ROA), operating cash flow, operating income, growth rate (GRO), Tobin's Q (Tobinq), equity concentration (Top1), enterprise size (Size), time to market (Age), money growth rate (M2), and economic growth rate (GDP) [26, 27, 73, 74]. Additionally, this study incorporates fixed effects for time and industry to mitigate the potential influence of industry-level disparities and latent time-related factors on enterprises.

**3.2.4 Grouping variables (var).**   To test the impact of the development of digital finance on the leverage ratio of different enterprises, the enterprises are grouped according to ownership (State), size (Siz), profitability (Pro), and leverage level. Where: state-owned enterprises consider the value of 1 and non-state-owned enterprises take a value of 0, enterprises larger than the average size of all enterprises are large-scale enterprises and take a value of 1; otherwise, they are small-scale enterprises and take a value of 0, enterprises whose operating income growth rate is below the average operating income growth rate of all enterprises are low-profit enterprises and take a value of 0; otherwise they are high-profit enterprises and take a value of 1, enterprises whose leverage ratios are higher than the average leverage ratio of all enterprises are highly leveraged enterprises, and they assume a value of 1. Otherwise, they are low-leverage enterprises, taking a value of 0. The specific definitions of each variable are shown in S1 Appendix.

**3.2.5 Mechanism variable.**   Financing constraint: by re-Ferring to Hadlock and Pierce [75], this paper employs the SA index to measure the financing constraints of enterprises. The specific calculation formula is as follows Where the mediation variables are financing cost (cost), financing constraint (K.Z) and non-systemic risk (risk), which are determined by the following formulas, cost = financial expenses divided by total operating income; K.Z comes from CSMAR database; risk reflects non-systemic risk, which is obtained by from the weekly yield of the stock and the weekly return of the market from CSMAR to regress the residuals and calculate the standard deviation to reflect the unsystematic risk profile.

**3.2.6 External economic impact of digital finance and leverage ratio.**   The study of the effects of the external economic environment on the relationship between digital finance development and corporate leverage ratio is crucial for understanding the potential dynamic between digital finance, the external economic context, and corporate financial structures. Digital finance, by offering alternative fund of sources and improving financial efficiency, has the potential to reduce corporate leverage ratios [76]. However, the external economic environment can change the extent to which this occurs [77]. In addition, the degree of financial supervision can influence enterprises' risk management decisions and the financial mechanism's overall solvency. By focusing on how factors such as the degree of marketization and financial supervision influence the relationship between digital finance and corporate leverage ratios, the study was conducted based on the previous model by Zhang et al [78], and Sagner [79]. This study further provides valuable insights into the mechanisms through which digital finance may mitigate leverage. Through rigorous analysis, this study not only explains the potential benefits of digital finance in reducing corporate leverage but also emphasizes the

**Table 2. Descriptive statistics.**

| Role | Variable | N | Mean | SD | Min | Max |
|---|---|---|---|---|---|---|
| Dependent variables [74, 80] | LLEV | 27,739 | 0.430 | 0.210 | 0.0100 | 1.760 |
| | SLEV | 27,739 | 0.340 | 0.180 | 0 | 1.400 |
| | LLEV | 27,739 | 0.0900 | 0.0800 | -0.0900 | 0.850 |
| Key independent variables [81] | DIF | 27,739 | 5.910 | 0.100 | 5.700 | 6.070 |
| Control variables [20, 26, 74] | FIX | 27,739 | 0.210 | 0.160 | 0 | 0.970 |
| | ROA | 27,739 | 0.0400 | 0.0800 | -1.860 | 0.880 |
| | Cashflow | 27,739 | 0.0500 | 0.0800 | -1.940 | 0.880 |
| | Growth | 27,739 | 5.750 | 813.2 | -1.310 | 130000 |
| | Tobinq | 27,739 | 2.160 | 2.870 | 0.150 | 259.1 |
| | Top1 | 27,739 | 0.350 | 0.150 | 0 | 0.900 |
| | Size | 27,739 | 22.19 | 1.330 | 15.58 | 28.64 |
| | Listage | 27,739 | 2.150 | 0.800 | 0 | 3.430 |
| | M2 | 27,739 | 11.40 | 3.350 | 6.990 | 18.95 |
| | GDP | 27,739 | 6.740 | 2 | 2.350 | 10.60 |

importance of regulatory environments and market conditions in shaping the effectiveness of digital financial solutions.

## 4. Analysis and results

### 4.1 Descriptive statistics

The descriptive statement presented in Table 2 show substantial variability in the leverage ratios among the sampled firms, with a mean leverage of 0.430 and a notable difference in preference for short-term debt (mean short-term leverage ratio of 0.34) as compared to long-term debt (mean long-term leverage ratio of 0.09). This disparity underscores the strategic financial decisions that enterprises make in the context of digital financial development, which averages a relatively high and low value of 5.91. When examining control variables, there is a right-skewed distribution in fixed assets, operating income growth, Tobin's Q, and economic growth, indicating that a minority of firms possess exceptionally high values in these areas. In contrast, variables such as listing time and currency growth rate display a left-skewed distribution. The congruence between average and median values for return on assets, operating cash flow, equity concentration, and enterprise size suggests these variables are normally distributed within the sample. This overview reveals the intricate implications of digital finance in shaping corporate financial leverage and highlights the need for firms to leverage flexible financial strategies that can accommodate the dynamic interplay of technological advancement and economic indicators.

### 4.2 Baseline regression

Table 3 presents the empirical findings derived from Model (1), indicating a significant and negative correlation between the development of digital finance and corporate leverage. Columns (2) and (4) display the outcomes of a univariate regression of digital financial development while controlling for industry and time-fixed effects. The inclusion of columns (2)-(1) in the regression demonstrates that digital finance development markedly reduces corporate leverage ratios. Furthermore, Columns (3) and (4) disentangle short-term leverage (SLEV) and long-term leverage (LLEV) for regression based on the term structure variance in corporate leverage. The results indicate a noteworthy negative impact of digital finance development

**Table 3. Baseline regression results.**

| Variables | (1) LEV | (2) LEV | (3) SLEV | (4) LLEV |
|---|---|---|---|---|
| DIF | -0.213*** | -0.137*** | -0.086*** | -0.050*** |
| | (-17.668) | (-13.661) | (-8.997) | (-9.225) |
| FIX | | 0.084*** | -0.035*** | 0.119*** |
| | | (9.938) | (-4.353) | (23.750) |
| ROA | | (9.938) | (-4.353) | (23.750) |
| | | -0.744*** | -0.593*** | -0.152*** |
| Cashflow | | -0.127*** | -0.037** | -0.089*** |
| | | (-7.332) | (-2.181) | (-10.192) |
| Growth | | 0.000*** | 0.000*** | -0.000*** |
| | | (14.242) | (23.396) | (-4.117) |
| Tobinq | | (0.003*** | (0.003** | 0.001*** |
| | | (2.890) | (2.529) | (3.738) |
| Top1 | | -0.004 | 0.019*** | -0.023*** |
| | | (-0.587) | (2.981) | (-6.409) |
| Size | | 0.069*** | 0.040*** | 0.029*** |
| | | (63.145) | (37.592) | (56.481) |
| Listage | | 0.030*** | 0.030*** | 0.001 |
| | | (19.426) | (20.261) | (0.709) |
| M2 | | 0.007*** | 0.006*** | 0.002*** |
| | | (10.557) | (8.597) | (4.501) |
| GDP | | 0.003*** | 0.003*** | 0.000 |
| | | (3.585) | (3.755) | (0.046) |
| Cons | 1.672*** | -0.479*** | -0.172*** | -0.313*** |
| | (23.142) | (-7.424) | (-2.789) | (-9.232) |
| Time effect | Yes | Yes | Yes | Yes |
| Industry effect | Yes | Yes | Yes | Yes |
| No. of obs. | 27,739 | 27,739 | 27,739 | 27,739 |
| R2_a | 0.2053 | 0.4620 | 0.3226 | 0.4230 |

Note

\*\*\*, \*\*, and \* represent 1%, 5%, and 10% statistical significance, respectively; t-values are in parentheses; standard errors are adjusted based on firm-level robust clustering, the same below.

on both short-term and long-term corporate leverage. In an economic context, a one-standard-deviation increase in digital finance development corresponds to a 0.137% decline in the sample standard deviation of the total corporate leverage ratio. Additionally, it is associated with 0.086% and 0.05% declines in the sample standard deviations of short-term and long-term leverage ratios, respectively. Control variables in Column (2) reveal that the proportion of fixed assets demonstrates a significant positive correlation with the corporate leverage ratio. This suggests that a higher proportion of fixed assets strengthens an enterprise's borrowing ability and facilitates more access to mortgage loans, elevating the leverage ratio. Return on assets exhibits a strong negative correlation, implying that a high asset utilization ratio increases internal funds, enhancing solvency and reducing the corporate leverage ratio. Operating cash flow is positively and significantly correlated with the corporate leverage ratio, indicating that increased operating cash flow alleviates financing constraints and provides more funds for debt repayment and operations, leading to a lower leverage ratio. Conversely, operating income growth shows a significantly positive correlation with the corporate leverage ratio, suggesting that firms with expanding incomes may leverage debt to fuel further growth. Equity concentration negatively correlates with the enterprise's leverage ratio, pointing to a lower debt level in enterprises with higher governance levels. Larger enterprises with more collateral

find it easier to secure credit, leading to higher debt levels. A longer time since going public facilitates more open access to credit markets, resulting in higher leverage. Lastly, a higher economic growth rate correlates positively with leverage, reflecting the ease of obtaining credit in a favorable macroeconomic environment for enterprises. These macroeconomic indicators underscore that a robust economic backdrop facilitates reduced debt levels for enterprises.

## 4.3 Robustness tests

**4.3.1 Endogenous test.**  Table 4 regression results presented that the crucial insights into the relationship between digital finance development and corporate leverage ratios. Across all four models, the coefficient for digital finance development (DIF) consistently demonstrates a statistical significance level of 1%, indicating a robust negative association with corporate leverage. Specifically, in the case of the dependent variable LEV (Corporate Leverage Ratio), the coefficient in Model (1) is -0.193, suggesting that for every unit increase in digital finance development, the corporate leverage ratio reduced by 0.193 units. This negative impact persists in Models (2), (3), and (4) for different specifications of the dependent variable, including overall leverage (LEV), short-term leverage (SLEV), and long-term leverage (LLEV). The statistical significance of the coefficients is supported by the remarkably high t-statistics, ranging from -6.910 to -13.774, emphasizing the reliability of the results. Moreover, the control variables, time effects, and industry effects consistently play a role in shaping the corporate leverage ratios, contributing to the overall explanatory power of the models. The large sample size (No. of Obs. = 19,288) enhances the reliability and generalizability of our findings. The R2_a values, though relatively low, indicate a proportionate variance in corporate leverage ratios explained by the models, with model (2) having the highest R2_a at 0.3206. The regression results reaffirm a robust and statistically significant negative relationship between digital finance development and corporate leverage ratios, providing valuable insights into the impact of digital finance on the financial structures of enterprises over both short- and long-term leverages.

**4.3.2 Sub-sample test.**   The changes in the financial environment will have a particular impact on the development of digital finance; ignoring these changes may affect the conclusions of the above research. Therefore, to exclude the impact of the financial events in China in 2015 on the financial environment, a sub-sample from 2011 to 2020 (except 2015) was used for regression as mentioned in Table 5. The results show that the above conclusions are robust.

**4.3.3 Introducing lagged terms.**   Previous periods' corporate leverage ratio may impact the current period's corporate leverage ratio. Therefore, to avoid omitting the corporate leverage ratio in prior periods and affecting the above research conclusion, the one-year lag of the corporate leverage ratio is introduced into regression (1). The results in Table 6 show that the research conclusions are robust.

**Table 4. Regression results.**

| Variables | (1) LEV | (2) LEV | (3) SLEV | (4) LLEV |
|---|---|---|---|---|
| DIF | -0.193*** (-13.774) | -0.142*** (-12.246) | -0.097*** (-8.734) | -0.045*** (-6.910) |
| Control variable | Yes | Yes | Yes | Yes |
| Time effect | Yes | Yes | Yes | Yes |
| Industry effect | Yes | Yes | Yes | Yes |
| No. of Obs. | 19,288 | 19,288 | 19,288 | 19,288 |
| R2_a | 0.0058 | 0.3206 | 0.1849 | 0.1922 |

**Table 5. Subsample (excluding 2015) regression results.**

| Variables | (1) LEV | (2) LEV | (3) SLEV | (4) LLEV |
|---|---|---|---|---|
| DIF | -0.212*** (-16.750) | -0.135*** (-12.846) | -0.086*** (-8.615) | -0.048*** (-8.435) |
| Cons | 1.670*** (22.043) | -0.473*** (-6.996) | -0.155** (-2.408) | -0.324*** (-9.138) |
| Control variable | Yes | Yes | Yes | Yes |
| Time effect | Yes | Yes | Yes | Yes |
| Industry effect | Yes | Yes | Yes | Yes |
| No. of Obs. | 25,410 | 25,410 | 25,410 | 25,410 |
| R2_a | 0.2032 | 0.4618 | 0.3231 | 0.4196 |

**4.3.4 Quantile regression.** To study how the impact of digital finance development is different under various leverage ratio levels and to conduct further robustness check, this paper performed quantile regression at 25%, 50%, and 75%, respectively. It can be seen from Table 7 that DIF coefficients under the three percentiles are always significantly negative, and the coefficients gradually become smaller as the percentiles increase. The results show that the above research conclusions are robust.

## 4.4 Heterogeneity

Since China is a vast country with huge differences in natural geographic conditions and economic development levels in different regions, the influence of digital financial development on corporate leverage ratios to the achievement of distinct enterprises may vary significantly. Through heterogeneity analysis, this study can inspect the causality between the achievement of corporate leverage ratios and digital financial inclusion more thoroughly to promote sustainable corporate financial development.

As mentioned above, digital finance can reduce enterprises' corporate ratios. The sampled enterprises are further into sub-grouped such as firm ownership (SOE), scale (SCA), profitability (PRO) and leverage level (HL); the relationship between digital finance development and these enterprises is created and estimated based on model (1) where Column (1) of Table 8 the results of grouping by ownership (SOE) show that the relationship between digital finance

**Table 6. Explanatory variables.**

| Variables | (1) LEV | (2) LEV | (3) SLEV | (4) LLEV |
|---|---|---|---|---|
| $LEV_{t-1}$ | 0.904*** (256.481) | 0.851*** (190.015) | | |
| $SLEV_{t-1}$ | | | 0.840*** (177.491) | |
| $LLEV_{t-1}$ | | | | 0.754*** (101.674) |
| DIF | -0.018*** (-3.098) | -0.019*** (-3.494) | -0.017*** (-3.031) | -0.008** (-2.058) |
| Cons | 0.151*** (4.372) | -0.097*** (-2.996) | -0.034 (-1.025) | -0.096*** (-3.871) |
| Control variable | Yes | Yes | Yes | Yes |
| Time effect | Yes | Yes | Yes | Yes |
| Industry effect | Yes | Yes | Yes | Yes |
| No. of Obs. | 23,337 | 23,337 | 23,337 | 23,337 |
| R2_a | 0.8472 | 0.8680 | 0.8130 | 0.7426 |

**Table 7. Quantile regression.**

| Variables | 25th percentile (1) LEV | 50th Percentile (2) LEV | 75th Percentile (3) LEV |
|---|---|---|---|
| DIF | -0.132*** (-9.20) | -0.136*** (-12.85) | -0.141*** (-10.22) |
| Control variable | Yes | Yes | Yes |
| Time effect | Yes | Yes | Yes |
| Industry effect | Yes | Yes | Yes |
| No. of Obs. | 27,737 | 27,737 | 27,737 |

development on the leverage ratio of non-state-owned enterprises is -0.135, while the regression coefficient of the relationship between digital finance development and the leverage ratio of state-owned enterprises (SOE) is -0.137, which is significant at the 1% level. The results indicate that digital finance development has a relatively high adverse effect on the leverage ratio of non-state-owned enterprises. Column (2) shows the results obtained by grouping regression according to the firm's size. It shows that for small-size enterprises, the impact of digital finance on the corporate leverage ratio has a coefficient of -0.137, while for large-scale enterprises, the regression coefficient is -0.137, which are significant at a 1% level. This shows that compared with large-scale enterprises, digital finance development has a relatively large and negative effect on the leverage ratio of small-scale enterprises. Column (3) shows that the regression according to enterprise profitability grouping, also compared with high-profit enterprises, the negative effect of digital financial development on the leverage ratio of low-profit enterprises is relatively large, but the difference is not significant. Similarly, in Column (4), the result indicates grouping regression according to the leverage level of enterprises, which shows that the negative impact of digital financial development on the leverage ratio of low-leverage enterprises is relatively large compared with high-leverage enterprises. The digital finance development on leverage ratios across different types of enterprises. By implementing the suggested recommendations, policymakers and financial institutions can strive to promote financial stability and sustainable growth for state-owned and non-state-owned enterprises,

**Table 8. Heterogeneity test ratio.**

| Variables | (1) LEV | (2) LEV | (3) LEV | (4) LEV |
|---|---|---|---|---|
| DIF | -0.135*** (-13.493) | -0.137*** (-13.690) | -0.137*** (-13.660) | -0.137*** (-13.659) |
| SOE*DIF | 0.003*** (6.951) | | | |
| SCA*DIF | | 0.001*** (2.700) | | |
| PRO*DIF | | | 0.002 (1.025) | |
| HL*DIF | | | | 0.001** (2.540) |
| Cons | -0.462*** (-7.155) | -0.477*** (-7.387) | -0.479*** (-7.423) | -0.478*** (-7.405) |
| Control variable | Yes | Yes | Yes | Yes |
| Time effect | Yes | Yes | Yes | Yes |
| Industry effect | Yes | Yes | Yes | Yes |
| No. of Obs. | 27,739 | 27,739 | 27,739 | 27,739 |
| R2_a | 0.4630 | 0.4621 | 0.4620 | 0.4621 |

small-scale and large-scale enterprises, as well as low-profit and both low and high-leverage enterprises. These targeted measures aim to facilitate better leverage management and capitalize on the opportunities presented by digital finance, ultimately contributing to enterprises' overall resilience and competitiveness in an evolving financial landscape.

Overall, the results in Table 8 shows that the development of digital finance can significantly reduce the corporate leverage ratio and that the impact of digital finance development on the corporate leverage ratio is heterogeneous due to the different characteristics of enterprises. Digital finance development has a more significant deleveraging effect on private, small-scale, and low-leverage enterprises, suggesting that H2 cannot be rejected.

## 5. Mechanism analysis

Based on existing literature analysis, this study addresses the impact of digital finance development on corporate financial leverage ratios. It declares that digital financial inclusion can mitigate operational risk and alleviate financing constraints. Therefore, this study investigates the operation and financing effects of digital transformation. Furthermore, developing digital finance can improve firms' decision-making processes by facilitating access to quality information. On the other hand, it can help enterprise quality management and encourage firm management productivity. These factors collectively reduce firms' volatility and minimize the likelihood of financial defaults. Based on the model (1), we adopted the following basic panel specification to examine the development of digital finance on corporate leverage ratios [82, 83]; the mediation effect model is constructed as follows: The study shows that digital finance development can significantly reduce the corporate leverage ratio and significantly deleverage private enterprises, small-scale enterprises, and low-leverage enterprises. Combined with the above analysis, further testing has been conducted to determine whether the impact of digital finance development on corporate leverage will be conveyed by reducing financing costs, alleviating financing constraints, and strengthening risk-taking.

$$lev_{it} = \lambda_0 + \lambda_1 dfi_{it} + \sum_j \varphi_j cv_{jit} + \sum_i \delta_i Industry_i + \sum_t \rho_t Year_t + \varepsilon_{it} \tag{2}$$

$$x_{it} = \alpha_0 + \beta_1 dfi_{it} + \sum_j \varphi_j cv_{jit} + \sum_i \delta_i Industry_i + \sum_t \rho_t Year_t + \varepsilon_{it} \tag{3}$$

$$lev_{it} = \varphi_0 + \beta_2 dfi_{it} + \beta_3 X_{it} + \sum_j \varphi_j cv_{jit} + \sum_i \delta_i Industry_i + \sum_t \rho_t Year_t + \varepsilon_{it} \tag{4}$$

The mediation variables are financing cost (cost), financing constraint (K.Z), and non-systemic risk (risk), which are determined by the following formulas: cost = financial expenses divided by total operating income; K.Z comes from CSMAR database; risk reflects non-systemic risk, which is obtained by from the weekly yield of the stock and the weekly return of the market from CSMAR to regress the residuals and calculate the standard deviation to reflect the unsystematic risk profile.

Table 9 Using the intermediary effect model to analyze the mechanism of digital financial development on corporate leverage ratio. Columns (1) and (4) of the results indicate the testing financing cost of intermediary variables. Specifically, Column (0) results show that digital financial development can significantly reduce corporate leverage. Where Column (1) shows that digital finance development is negatively correlated with the financing cost of enterprises at 1% level of significance, that means the development of digital finance helps reduce enterprises' financing cost. Column (4) also considers the impact of digital finance development and financing costs on corporate leverage ratios, from which can be seen that digital financial development is significantly negatively correlated with corporate leverage, and financing costs

**Table 9. Mechanism analysis.**

| Variables | (0) LEV | (1) LEV | (2) LEV | (3) LEV | (4) LEV |
|---|---|---|---|---|---|
| DIF | -0.137*** (-13.661) | -0.011** (-2.293) | -1.173*** (-11.171) | -0.002* (-1.648) | -0.076*** (-9.196) |
| Cost | | | | | 0.014*** (2.611) |
| KZ | | | | | 0.055*** (79.909) |
| Risk | | | | | 0.572*** (10.873) |
| Cons | -0.479*** (-7.424) | 0.019 (0.602) | 3.552*** (4.556) | 0.145*** (21.403) | -0.725*** (-13.568) |
| Control variable | | Yes | Yes | Yes | Yes |
| Time effect | | Yes | Yes | Yes | Yes |
| Industry effect | | Yes | Yes | Yes | Yes |
| No. of Obs. | 27,739 | 27,737 | 27,737 | 27,257 | 27,257 |
| R2_a | 0.4620 | 0.0086 | 0.4571 | 0.3195 | 0.6470 |

are significantly positively correlated with corporate leverage. This is because the higher financing cost can raise the cost of the enterprise's debt, and the lower income and the less return can affect the debt repayment, resulting in a higher corporate leverage ratio. Reducing financing costs is an intermediary channel for developing digital finance to act on the leverage ratio of enterprises. Meanwhile, the Bootstrap test of the mediation effect shows that the indirect effect and confidence interval does not include 0, indicating that the mediation effect is valid.

Table 9 In columns (2) and (4), the results indicate testing the intermediary variable, the financing constraint (K.Z). As seen in Column (2), there is a significant negative correlation between the development of digital finance and financing constraints. The result suggests that digital finance development can significantly reduce enterprises' financing constraints. Column (4) indicates the regression result of the relationship between digital financial development, corporate financing constraints, and corporate leverage, which shows that digital finance development negatively correlates with the corporate leverage ratio at the 1% significance level. The financing constraints of enterprises are positively correlated with the corporate leverage ratio at the 1% significance level. The results also indicate that when an enterprise is focused on more significant financing constraints, it is more likely that the funds required for the enterprise's production and operations will be difficult to guarantee, resulting in a decrease in the surplus funds that can be used to repay debts and an increase in the leverage ratio. From columns (2) and (4), easing financing constraints is an intermediary factor for digital finance development to affect the corporate leverage ratio. Moreover, the Bootstrap test shows a significant mediation effect at 1% significance level, as shown in Table 10.

Similarly, columns (3) and (4) are intermediary variables indicating non-systemic risk. In Column (3), the operating risk of digital finance development is significantly negatively correlated at the 1% level of significance, suggesting that digital finance development can reduce non-systemic risks. Column (4) shows the simultaneous relationship between digital financial development, corporate operating risk, and corporate leverage ratio. The result shows that the development of digital finance is negatively correlated with corporate leverage at the level of 1%, and risk assumption is positively correlated with the leverage ratio of enterprises at the level of 1%. This is because when the non-systemic risk of the enterprise increases, its operating income will decrease, and the surplus used to repay the debt will also decrease, increasing

**Table 10. Bootstrap test of mediating effect.**

| | Path | Coefficient | S.E. | Bootstrap 95% CI | |
|---|---|---|---|---|---|
| | | | | Lower bound | Upper bound |
| Cost | Indirect effect | -.0001* | .0001 | -.0006 | -0.0000 |
| | Direct effect | -.1412*** | .0081 | -.1538 | -.1162 |
| K.Z. | Indirect effect | -.0647*** | .0047 | -.0712 | -.0565 |
| | Direct effect | -.0766*** | .0076 | -.0844 | -.0593 |
| Risk | Indirect effect | -.0020** | .0011 | -.0039 | -.0000 |
| | Direct effect | -.1373*** | .0080 | -.1535 | -.1177 |

Notes: S.E. = standard error; CI = confidence interval; the above confidence intervals are bias-corrected confidence intervals

the leverage ratio of the enterprise. Reducing enterprises' unsystematic risk is an intermediary channel through which digital finance development affects corporate leverage ratios. Meanwhile, the Bootstrap test shows that this mediation effect is significant at the 1% level. In summary, reducing financing costs, easing financing constraints and weakening non-systemic risks are the factors through which digital finance development can reduce corporate leverage.

# 6. External economic impact of digital finance and leverage ratio

## 6.1 The degree of marketization

Table 11 presents the impact of digital finance development on corporate leverage in both low and high degrees of marketization. In the low degree of marketization column (1), the coefficient for DIF is -0.149, indicating a negative association between increased digital finance development and corporate leverage reduction in regions with low marketization levels. Additionally, the coefficient of -0.426 suggests a significant negative impact on leverage even when controlling for other factors, emphasizing the importance of digital finance in leverage reduction. Similarly, in the high degree of marketization column (2), the DIF coefficient value of -0.144 mirrors the findings in low marketization scenarios, indicating that increased digital finance development in highly marketized regions significantly reduces corporate leverage. This underscores the broad impact of digital finance on economic and social development, with its effects on corporate leverage varying depending on enterprise characteristics and the economic environment. Columns (1) and (2) also provide insights from group tests using the China Marketization Index sourced from the Wind database. Enterprises located in regions with marketization index values above the median are categorized as high marketization areas,

**Table 11. External economic impact of digital finance and leverage ratio.**

| | Low degree of marketization (1) Lev | High degree of marketization (2) Lev | Loose supervision (3) Lev | Strict supervision (4) Lev |
|---|---|---|---|---|
| DIF | -0.149*** (-9.033) | -0.144*** (-8.150) | -0.147*** (-9.356) | -0.136*** (-10.686) |
| Cons | -0.426*** (-3.747) | -0.564*** (-5.037) | -0.542*** (-5.353) | -0.411*** (-5.091) |
| Control variable | Yes | Yes | Yes | Yes |
| Time effect | Yes | Yes | Yes | Yes |
| Industry effect | Yes | Yes | Yes | Yes |
| No. of Obs. | 9,598 | 9,313 | 9,989 | 17,748 |
| R2_a | 0.4827 | 0.4759 | 0.5517 | 0.4344 |

while those below are considered low marketization areas. The results from Table 11 indicate that in high marketization areas, the negative impact of digital finance development on corporate leverage is less pronounced compared to low marketization areas. This suggests that continued promotion of marketization enhances the stability of leverage ratios, underscoring the pivotal role of marketization in driving economic growth.

## 6.2 Degree of financial supervision

Table 11 shows in Column 3, loose supervision represents the DIF coefficient -0.147 in an environment of loose supervision. The coefficient signifies a significant negative relationship between digital finance development and corporate leverage. The coefficient is -0.542; the constant term continues to harm leverage, while the coefficient is 0.564. The constant term remains negatively significant, implying a consistent impact on leverage. In Column 4, Strict Supervision -0.136 indicates that. Even under strict supervision, increased digital finance development correlates with a significant reduction in corporate leverage. The coefficient value -0.411 represents the constant term, which is negatively significant, indicating a consistent impact on leverage. The regulation of digital finance will not only affect its development process but may also affect its relationship with corporate leverage. To test whether the degree of financial supervision affects the relationship between digital finance development and corporate leverage ratio, following [84, 85]. The columns (3)-(4) of Table 11 indicate the negative impact of digital financial development on corporate leverage ratio during the period of strict financial supervision, this shows that appropriate deregulation of digital finance will help to improve the deleveraging effect.

## 7. Conclusions and policy recommendations

### 7.1 Conclusions

Based on a sample of 2011–2020 A-share non-financial listed companies in China, this paper examines the impact and mechanism of digital finance development on corporate leverage. The following conclusions can be drawn from the research findings: Firstly, digital finance development shows a significant negative correlation with the corporate leverage ratio. Moreover, after considering the difference in the term structure of corporate leverage ratio, it found that digital finance development exhibits a significant negative correlation with both short-term and long-term corporate leverage ratios. This suggests that digital finance development contributes to reducing the leverage ratio of enterprises, be it short-term or long-term leverage. Secondly, the negative impact of digital financial development on corporate leverage is heterogeneous due to the different characteristics of enterprises, including state-owned and non-state-owned enterprises, large and small-scale enterprises, and high-leverage and low-leverage enterprises. Third, Mechanism analysis demonstrates that the development of digital finance can reduce the corporate leverage ratio by lowering financing costs, easing financing constraints, and weakening non-systemic risks. Finally, Research on the external economic environment suggests that actively promoting the marketisation process and appropriately strengthening financial supervision will help enhance the positive impact of digital financial development on enterprise deleveraging.

This study empirically analyses the relationship between the development of digital finance and corporate leverage ratios within the existing theoretical framework and data availability. However, due to the initial stage of corporate leverage ratio disclosure and the lack of leverage data by many listed firms, the available data for this study has some limitations. In addition, we examine the mechanism through which digital finance affects corporate leverage ratios. There may be another possible technique through which digital finance may impact corporate

leverage performance. Future studies can be conducted from corporate financing performance, internal audit mechanisms, and digital finance perspectives. This study focuses on the improvement that digital finance brings to enterprises' leverage ratio performance; it would also be worthwhile to evaluate the positive impact of digital finance on firms' operational performance. All these highlighted problems above are worth recommending in depth for future research.

## 7.2 Policy recommendations

Based on the above analysis, this paper puts forward the following policy suggestions: First, increase policy support and actively promote the marketization process to improve the deleveraging effect of digital finance. The development of digital finance can not only reduce the leverage ratio of enterprises, but also the impact of deleveraging is affected by the degree of marketization. Therefore, increasing policy support for the development of digital finance, establishing and improving the infrastructure for the development of digital finance, and actively promoting the marketisation process will help to play the deleveraging role of digital finance better. Second, innovate digital financial supervision and guide the healthy development of digital finance to strengthen the positive effect of digital finance on corporate deleveraging. The development time of digital finance is short, and its development model is not yet mature. Financial supervision needs to be adequately strengthened to regulate and guide its healthy development. Therefore, to effectively play the role of the development of digital finance in deleveraging enterprises, the regulatory authorities should innovate the financial supervision model while strengthening the supervision of digital finance and preventing its potential risks to ensure the healthy development of digital finance and strengthen its positive effect. Third, State-owned enterprises in China often have access to preferential lending rates and support from the government, which can result in higher leverage ratios compared to non-state-owned enterprises. On the other hand, non-state-owned enterprises may face more stringent lending conditions and higher borrowing costs, leading to lower leverage ratios. This study further policy recommends implementing a standardized and transparent framework for determining corporate leverage ratios applicable to state-owned and non-state-owned enterprises. This framework should consider different industries' specific characteristics and risks while ensuring consistency in measuring leverage ratios across enterprises. Finally, developing digital finance is an opportunity to promote digital links between financial institutions and enterprises so that enterprises can actively participate in deleveraging. Information has become a key element in the development of modern enterprises. By strengthening the digital connection between financial institutions and enterprises, financial institutions can timely and effectively understand the operating conditions of enterprises. On the other hand, it can improve information transparency among economic agents and enable enterprises to actively participate in deleveraging and control the leverage ratio within a reasonable range.

## Supporting information

**S1 File.**
(ZIP)

**S1 Appendix. Variable concept and definitions.**
(DOCX)

## Author Contributions

**Data curation:** Liu Junqi, Sher Abbas, Liu Rongbing.

**Formal analysis:** Sher Abbas, Liu Rongbing, Najabat Ali.

**Investigation:** Sher Abbas, Najabat Ali.

**Methodology:** Liu Junqi, Sher Abbas, Liu Rongbing, Najabat Ali.

**Software:** Sher Abbas, Liu Rongbing.

**Supervision:** Liu Junqi.

**Validation:** Liu Junqi, Sher Abbas, Liu Rongbing, Najabat Ali.

**Visualization:** Sher Abbas.

**Writing – original draft:** Liu Junqi, Sher Abbas, Liu Rongbing, Najabat Ali.

**Writing – review & editing:** Liu Junqi, Sher Abbas, Liu Rongbing, Najabat Ali.

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
