## [Decision Letter · Decision Letter 0]

23 Jan 2024

PONE-D-23-44191­­­­­­­­­­The Impact of Digital Financial Development on Corporate Leverage Ratio: The Case of A-Share Listed Non-Financial Enterprises in China's Shanghai and Shenzhen Stock ExchangesPLOS ONE

Dear Dr. Abbas,

Thank you for submitting your manuscript to PLOS ONE. After careful consideration, we feel that it has merit but does not fully meet PLOS ONE’s publication criteria as it currently stands. Therefore, we invite you to submit a revised version of the manuscript that addresses the points raised during the review process.

We look forward to receiving your revised manuscript.

Kind regards,

Ricky Chee Jiun Chia

Academic Editor

PLOS ONE

Journal Requirements:

Reviewers' comments:

Reviewer's Responses to Questions

**Comments to the Author**

1. Is the manuscript technically sound, and do the data support the conclusions?

Reviewer #1: Yes

Reviewer #2: Yes

Reviewer #3: Partly

2. Has the statistical analysis been performed appropriately and rigorously? 

Reviewer #1: Yes

Reviewer #2: Yes

Reviewer #3: Yes

3. Have the authors made all data underlying the findings in their manuscript fully available?

Reviewer #1: No

Reviewer #2: Yes

Reviewer #3: No

4. Is the manuscript presented in an intelligible fashion and written in standard English?

Reviewer #1: Yes

Reviewer #2: No

Reviewer #3: Yes

5. Review Comments to the Author

Reviewer #1: 1. A literature review of relevant research methodologies is missing from the research synthesis, and it is recommended that it be supplemented.

2. In the manuscript, "H2: The negative impact of digital finance development on corporate leverage ratios is different among state-owned and non-state-owned enterprises." should read H3.

3. This study focuses on the heterogeneity of the impact of the development of digital finance on the leverage ratios of state-owned and non-State-owned enterprises, and large and small enterprises, but the "Policy recommendations" section does not explicitly propose differentiated regulatory strategies for different types of enterprises.

4. Please identify the software platform used for the calculation of the econometric model and provide further details on the rationale for the econometric model.

5. It is recommended that a technology roadmap be added to visualize the framework of the paper.

6. Please explain the need for the study "6. The impact of the external economic environment on the relationship between digital finance development and corporate leverage ratio" in the manuscript, and clarify the role of the research in the sixth part of the article in the overall context.

Reviewer #2: 1、Reading the abstract and introduction section, readers may find it hard to understand the study aims and contributions of the manuscript. The interesting points are vague. The current abstract and introduction section cannot attract readers' interests. The authors need to explain what is interesting and important.

2、Abbreviations that appear for the first time need to be elaborated

3、Literature review：the main ideas of the literature should be pointed out. The citation presentation and citation format of the literature review need to be changed，please read some related papers for the writing of literature review.

4、H1 is not even a complete sentence.

 5、Some opinions showed up in the paper should add relevant literature to support the point of view to make the further discussions be more convinced for the readers for example: Page 15 ”The development of digital finance can not only reduce the degree of financing discrimination suffered by small-scale enterprises, increase their sources of financing, but also stabilize the intermediary income of large-scale enterprises, thereby reducing the leverage ratio of the two. However, compared with small-scale enterprises with relatively low leverage levels, the development of digital finance may significantly impact the leverage ratio of large-scale enterprises with higher leverage levels.” )

6、The basis for the selection of control variables and the corresponding references are not mentioned.

7、The data collected only till 2020,would it be possible to retain updated data?

8、A mistake shows up in page 20“The control variables in column (2)-(2) show that …..”

9、Table 3, the directions of the coefficients of some control variables are positive or negative in different column, please indicate the specified reason of each variable.

10、The result showed in table 4 is one stage or second stage result? The reason to choose that instrumental variables should be indicated in the main text.

11、The potential reason of the heterogeneity should be explained in detail.

12、Mechanism: Insufficient elaboration of theoretical analyses for mechanism analysis

13、conclusion and recommendation need modification and the authors are suggested to advise such type of suggestion that is practical attainable.

14、Indicate direction for future research and propose some improvements regarding digital financial development on corporate leverage ratio.”

15、Language: It is noted that your manuscript needs careful polishing by someone with expertise in technical English editing so that the goals and results of the study are clear to the reader.

16、Font size and line spacing need to be consistent，Each part heading is too long

Reviewer #3: Thank you for the opportunity to review this paper, which deals with the topical issue of the impact of digital finance on enterprise leverage for the case of China, 2011-2020. The data, methodology, and research topic are generally reasonable and necessary to investigate. However, there are several points regarding literature, methodology, results, and discussion, which I find necessary to rethink and probably re-conceptualize in the paper. These points are listed below:

- The motivation and reasoning for the study are clearly stated in the introduction. However, the literature review could develop the motivation of the study in greater detail. Thus, the need to estimate digital finance's impact on the leverage ratio before hypothesis 1 is introduced could be presented in more detail.

- Additionally, Hypothesis 1 has to be reformulated. The formulation “There exists the significant impact of digital financial development on corporate leverage ratios” would probably be better.

- The reasoning for hypotheses 2 and 3 should include the particularity of the case of China.

- The reasoning for hypothesis 3 is a little mixed up. If the hypothesis focuses on state vs. non-state enterprises, the argumentation for large vs. small enterprises should not be included in the description. On the other side, based on the presentation of the results, an additional hypothesis regarding firm size could be introduced.

- For the case of the research design, the reasoning for filtering the data could be explained in greater detail. For example, why specifically did you choose to delete ST firms and financial and real estate segments?

- Additionally, the results of the tests of the model assumptions need to be presented in the appendix.

- It would be helpful if you included the source of each of your variables in Table 1.

- Probably, already when presenting descriptive results some possible interpretation for specific distributions could be provided (chapter 4.1).

- It can be suggested that some of the robustness checks can be moved to the appendix to improve the readability of the paper.

- It is unclear where some of the results in chapter 4.4 are presented, as I could not see them in Table 8. These relate to the following indicators “…development and the leverage ratio of state-owned enterprises is -0.133, which is also significant at 1% level”, “…while for large-scale enterprises, the regression coefficient is -0.136, which is also significant at a 1% level”. Please, re-check this.

- For the case of chapters 5 and 6, the reasoning for introducing these analyses is not clear, as it is not reflected in the theoretical part of the paper and the hypotheses. Please, reconsider the necessity of their inclusion in the main part of the paper, the reasoning for their inclusion and probably add the relevant hypotheses/reasoning in the theoretical part. In the current state the theoretical and the analytical parts are not completely aligned.

- Limitations of the study as well as the embedding of the results in current theoretical discussion should be added to chapter 7.

- Generally, it is necessary to re-read the paper for typos and formatting errors, as some could be identified in the paper (e.g., words “Table 9” at the beginning of two paragraphs on page 26).

6. PLOS authors have the option to publish the peer review history of their article (what does this mean?). If published, this will include your full peer review and any attached files.

Reviewer #1: No

Reviewer #2: No

Reviewer #3: No

---

## [Editor Report · Decision Letter 1]

16 Apr 2024

­­­­­­­­­­The Impact of Digital Financial Development on Corporate Leverage Ratio: The Case of A-Share Listed Non-Financial Enterprises in China's Shanghai and Shenzhen Stock Exchanges

PONE-D-23-44191R1

Dear Dr. Sher Abbas,

We’re pleased to inform you that your manuscript has been judged scientifically suitable for publication and will be formally accepted for publication once it meets all outstanding technical requirements.

Kind regards,

Ricky Chee Jiun Chia

Academic Editor

PLOS ONE
---

## [Editor Report · Acceptance letter]

2 Aug 2024

PONE-D-23-44191R1 

PLOS ONE

Dear Dr. Abbas, 

I'm pleased to inform you that your manuscript has been deemed suitable for publication in PLOS ONE. Congratulations! Your manuscript is now being handed over to our production team.

Kind regards, 

on behalf of

Dr. Ricky Chee Jiun Chia 

Academic Editor

PLOS ONE